# Photodynamic Inactivation Enhances Antibiotic Efficacy Without Affecting Drug Stability: Insights into Photosensitizer–Antibiotic Combination Therapies

**DOI:** 10.3390/ijms262311267

**Published:** 2025-11-21

**Authors:** Rocío B. Acosta, Edgardo N. Durantini, Mariana B. Spesia

**Affiliations:** IDAS-CONICET, Departamento de Química, Facultad de Ciencias Exactas, Físico-Químicas y Naturales, Universidad Nacional de Río Cuarto, Ruta Nacional 36 Km 601, Río Cuarto X5804BYA, Córdoba, Argentina; racosta@exa.unrc.edu.ar (R.B.A.); edurantini@exa.unrc.edu.ar (E.N.D.)

**Keywords:** antimicrobial, antibiotics, bacteria, cationic porphyrin, combination therapies, *Escherichia coli*, photodynamic inactivation, *Staphylococcus aureus*

## Abstract

Photodynamic inactivation (PDI) represents a promising strategy to overcome bacterial resistance by combining light, oxygen, and a photosensitizer (PS) to generate reactive oxygen species (ROS) that damage essential cellular components. Combining PDI with conventional antibiotics (ATBs) may further enhance bacterial eradication through complementary mechanisms. In this study, the tetracationic 5,10,15,20-tetra(4-*N*,*N*,*N*-trimethylammoniophenyl)porphyrin (TMAP^4+^) was evaluated in combination with ATBs: ampicillin (AMP) and rifampicin (RIF) against *Staphylococcus aureus* and cephalexin (CFX) against *Escherichia coli*. The photostability of all agents was assessed under the experimental irradiation conditions, and no evidence of physical interaction between TMAP^4+^ and the ATBs was detected. AMP and CFX remained photostable, while RIF exhibited only minimal photodegradation under white light, confirming its stability during PDI treatments. The antimicrobial assays revealed that irradiation significantly enhanced the bactericidal activity of TMAP^4+^. When combined with ATBs, photoactivated TMAP^4+^ led to a pronounced reduction in the minimum inhibitory concentration (MIC) values of AMP and RIF for *S. aureus* and of CFX for *E. coli*, indicating additive effects. Growth curve analyses corroborated these results, showing delayed bacterial growth and decreased maximal optical densities in the combined treatments compared to single agents. Overall, these findings demonstrate that the photodynamic process can potentiate the antimicrobial effect of conventional ATBs without compromising their stability, supporting the potential of PS–ATB combination therapies as a valuable approach to improve antibacterial efficacy and mitigate ATB resistance.

## 1. Introduction

The global increase in bacterial resistance to antibiotics (ATBs) is one of the main threats to public health, as it compromises the effectiveness of conventional treatments for common infection [1]. The selective pressure exerted by the extensive use of antimicrobials has favored the emergence of multidrug-resistant strains, driving the search for alternative or complementary therapeutic strategies capable of restoring antimicrobial efficacy [2]. In this context, photodynamic inactivation (PDI) has emerged as a promising tool due to its mechanism of action, which is independent of traditional routes of antibiotic resistance [3].

PDI is based on the excitation of a photosensitizer (PS) by visible light in the presence of molecular oxygen, generating highly cytotoxic reactive oxygen species (ROS) [4]. These species can oxidize lipids, proteins, and nucleic acids, causing irreversible damage to multiple cellular targets [5,6]. Unlike ATBs, the action of PDI does not depend on a specific molecular site, which makes it difficult for bacteria to develop resistance [7]. However, the effectiveness of the therapy can be influenced by the structural characteristics of the bacteria. In particular, the presence of an outer membrane in Gram-negative bacteria acts as a barrier that limits the diffusion of negatively charged or hydrophilic molecules, including many PS, into the cytosol [8]. Among cationic PSs, substituted porphyrins have demonstrated high quantum efficiency in singlet oxygen generation and affinity for bacterial surfaces. In particular, the tetracationic derivative 5,10,15,20-tetrakis(4-*N*,*N*,*N*-trimethylammoniumphenyl) porphyrin (TMAP^4+^) has a singlet oxygen quantum yield of 0.77 in water [8] and has shown antimicrobial activity due to its ability to interact electrostatically with cell membranes and generate ROS under visible light [9].

One of the advantages of PDI is that it can be used in conjunction with other therapies. The combined use of PDI with ATBs has been proposed as a synergistic strategy capable of enhancing antimicrobial action and reducing the necessary concentrations of each agent, avoiding the side effects of some drugs and the ability to generate resistance [10,11]. Ampicillin (AMP), rifampicin (RIF), and cephalexin (CFX) are broad-spectrum ATB that act through distinct mechanisms. AMP and CFX belong to the β-lactam class and inhibit bacterial cell wall synthesis by binding to penicillin-binding proteins, leading to cell lysis. However, their efficacy can be limited by the production of β-lactamases and by reduced permeability in Gram-negative bacteria [12]. In contrast, RIF inhibits bacterial RNA polymerase, blocking transcription and protein synthesis, but its clinical use is often compromised by the rapid emergence of resistant mutants [13]. PDI can alter the permeability of the cell envelope, favoring the entry of the ATB [3,14], while certain β-lactams, like AMP or CFX, can facilitate PS penetration and increase intracellular ROS production [15,16]. In contrast, RIF could experience additive or synergistic effects of different origins, related to oxidative damage to genetic material or the simultaneous inhibition of RNA synthesis [17]. Therefore, the combination of these ATBs with PDI represents a rational strategy to potentiate antimicrobial efficacy through complementary mechanisms.

This study evaluated the antimicrobial activity of the cationic PS TMAP^4+^ in combination with ATBs exhibiting distinct mechanisms of against different bacterial models (AMP and RIF against *S. aureus*, and CFX against *E. coli*). To ensure that the observed antimicrobial activity resulted from the additive effects of the combined treatments, preliminary experiments were conducted to confirm the absence of interaction between the PS and each ATB. Furthermore, the influence of protein-rich medium conditions and visible light exposure was evaluated on their stability. Minimum inhibitory concentrations (MIC) and minimum bactericidal concentrations (MBC) were determined under dark conditions and after irradiation to assess the contribution of PDI to the overall efficacy of the combination therapies. In addition, bacterial growth curves were monitored to examine population dynamics in the presence of single and combined agents.

## 2. Results and Discussion

### 2.1. Interaction Between TMAP^4+^ and ATBs

To effectively combine two therapies with different mechanisms of action, it is essential to assess whether interactions between their components or with the experimental conditions may occur. Such interactions can induce changes in their activity and consequently determine the success or failure of the combination. For this reason, the possible interaction between the PS and the ATBs was initially evaluated. The fluorescence spectra of the PS were recorded in the presence of increasing concentrations of each ATB. Figure 1 show that the fluorescence intensity of TMAP^4+^ remained essentially unchanged, varying only within the instrumental error range. No systematic decrease was observed with increasing ATB concentration, and the ΔF = (F − F_0_) vs [ATB] plots lacked any linear correlation (Appendix A). These findings demonstrate the absence of interaction between TMAP^4+^ and any of the ATBs tested. This absence of a defined correlation confirms that the PS remains spectroscopically and functionally independent under the studied conditions.

### 2.2. MIC and MBC Determination

ATB MICs were first determined for each strain using the microdilution method [18]. The CFX MIC on *E. coli* was 8 μg/mL and the CBM was 16 μg/mL. The treatment of *S. aureus* with AMP obtained a MIC of 0.0625 μg/mL and with RIF 0.0078 μg/mL, while the MBC was 0.1250 and 0.0156 μg/mL, respectively. In all cases, the CBM corresponds to twice the concentration necessary to cause an inhibitory effect on the development of microorganisms (CIM). These values are consistent with the results suggested in the Clinical and Laboratory Standards Institute (CLSI) manual for reference strains [19].

Subsequently, the MIC and MBC of TMAP^4+^ were determined. The results, shown in Table 1, revealed that in dark conditions, both strains were inhibited at a concentration of 32 mM, indicating that at high concentrations, the PS exhibits intrinsic toxicity even without irradiation. This dark toxicity is mainly attributed to the strong affinity of TMAP^4+^ for bacteria envelopes. Electrostatic interactions between the positively charged porphyrin and the negatively charged bacterial surface can disrupt membrane permeability and integrity, even in absence of light-induced ROS generation [8]. This result was confirmed by determining the MBC, where it was observed that the same concentrations that inhibit the growth of both microorganisms are also bactericidal in the dark. However, after irradiation (30 min) the MIC of TMAP^4+^ for *S. aureus* was 64 times lower than the concentration needed to achieve the same effect in the dark and was significantly lower than that for *E. coli*. However, it can be observed that light-activated PS showed no significant differences in the inhibition of *E. coli* growth compared to the cultures in the dark. On the other hand, the value obtained for the post-irradiation MBC for *E. coli* coincides with the pre- and post-irradiation MIC value, which indicates that using 32 µM the eradication of bacteria occurs. For *S. aureus*, the concentration of TMAP^4+^ needed to eradicate (MBC post PDI) the cells after irradiation was 32 times higher than that necessary to inhibit (MIC post PDI) microbial growth. Other authors have reported similar findings, observing differences in the MICs values of *meso*-tetra(4-aminophenyl)porphine on *S. aureus* and *E. coli*. The concentration required to inhibit the growth of the Gram-negative strain was approximately 6 times higher than that required for the Gram-positive strain [20]. This may be due to differences in the composition of the structures coating Gram-positive and Gram-negative bacteria, as well as a higher amount of PS associated with *S. aureus* [8]. One of the biggest differences is the outer membrane found in Gram-negative bacteria, which acts as a barrier that limits the diffusion of PS into the cytosol [21].

### 2.3. Determination of MIC and MBC of PS + ATB at a Fixed Concentration of PS

The MIC previously determined for each of the agents separately were used as a reference to design the combined assays. In addition, the MIC and MBC values of the ATBs were corroborated individually after subjecting them to the PDI conditions (30 min irradiation), to exclude possible adverse effects of light on their antimicrobial activity (see *Photostability of ATBs* below). The results confirmed that all ATBs retained their antimicrobial action after this irradiation time. After applying the therapies together, the results obtained were analyzed (Table 2). For *S. aureus*, the fixed concentration of PS used was 0.5 μM. Under this condition, a significative decrease in the MIC of the ATBs obtained after irradiation was observed. Specifically, the MIC of AMP decreased 4-fold while the MIC of RIF has halved, compared to the respective MICs of these ATBs used individually (Appendix A). However, this enhanced effect was absent in cultures maintained in the dark. These findings highlight the role of PDI, indicating that the ROS generated from PS excitation potentiates ATB action. Likewise, the contribution of PDI in the determination of the MBC after irradiation was observed, since in both cases it was lower than the MBC determined in the dark. However, while the MBC for AMP was unchanged, the MBC for RIF was twice as high after irradiation compared to the ATBs used alone.

For *E. coli* cultures, two fixed concentrations of TMAP^4+^ were analyzed. First, a concentration of 32 μM was used to corroborate the previously observed eradication effect. The results obtained confirmed the complete elimination of microorganisms even at the lowest concentration of ATB tested. Afterward, PS was applied at half the concentration (16 μM). Under this condition, no differences were found between the MIC obtained before and after irradiation. However, a decrease of half of CFX MIC was observed compared to that obtained in the absence of PS (Appendix A). In addition, the concentration needed for complete eradication of *E. coli* was twice that required for growth inhibition. These results indicate that each therapy contributes separately to increasing antimicrobial activity, allowing lower concentrations of both agents.

### 2.4. Determination of MIC and MBC of PS + ATB at Different Concentrations of PS

The ability to inhibit bacterial growth was analyzed using varying concentrations of both agents. The results are detailed in Table 3. For the Gram-positive strain, the combination of AMP with irradiated-TMAP^4+^ resulted in a lower MIC than the agents applied individually (Appendix A). A similar potentiating effect was observed when RIF was joint to PDI. In addition, a significant difference in MIC values was observed before and after irradiation. This result indicates that PDI contributed to the antimicrobial effect. After irradiation, the MIC was consistently lower in tests combining PDI with both ATBs, confirming that the ATB + PDI combination enhances antimicrobial activity more than either therapy alone. Regarding the MBC, the values were, as expected, higher than MIC. There was a noticeable difference in MBC values between the dark and illuminated tests when AMP was used. However, this change was not observed with the RIF + PDI combination.

In the case of the Gram-negative strain, a decrease in the concentrations of both CFX and TMAP^4+^ required to inhibit microbial growth was also observed compared to those of each agent separately. However, the CIM was significant lower (2–1 μg/mL CFX + 8–4 μM TMAP^4+^) (Appendix A) than the concentrations needed for the fixed dose combination (4 μg/mL CFX + 32 μM TMAP^4+^) and even more than the agents used separately (8 μg/mL CFX + 32 μM TMAP^4+^). This represents a 4-fold decrease in the CIM of each agent against *E. coli*. Similarly, the joint MBC was 8 μg/mL CFX + 32 μM TMAP^4+^, signifying a reduction in the CBM of ATB necessary to achieve the same effect as when used alone. Therefore, the results suggest a mutual potentiation between the action of ATB and the PS. It has been proposed that the action of ATBs that target cell wall synthesis, such as AMP and CFX, could destabilize the cell envelope, thereby facilitating the action of PS. This is consistent with previous studies that have demonstrated similar potentiation between ATBs that inhibit cell wall synthesis and porphyrins on *E. coli* and *S. aureus* [15,22]. By contrast, RIF, which acts by inhibiting RNA synthesis, does not alter membrane permeability, which may explain the more modest enhancement observed with this drug. While an increasing effect could still occur if PDI facilitates the entry of ATB into the cell, the underlying mechanism is less clear [23].

### 2.5. Interaction Between TMAP^4+^ and BSA

The bacterial growth environment is a key factor determining the effectiveness of PDI-ATB combination therapies. Among its components, the nature and abundance of proteins can strongly affect the photoactivation efficiency of PSs. It is well established that cationic PSs can interact with medium proteins, potentially altering their availability, photophysical behaviour, and consequently, their photodynamic efficiency [24,25]. Considering that albumin is the major protein present in serum and one of the most abundant in biological systems, its interaction with TMAP^4+^ was investigated to simulate protein-rich environments and evaluate potential bioavailability issues. The results are shown in Figure 2. As depicted in Figure 2A, the fluorescence intensity of TMAP^4+^ decreased progressively with increasing BSA concentration, indicating a quenching effect associated with complex formation. The linear correlation observed between the F_0_/F ratio and the molar concentration of BSA (Figure 2B) confirmed the occurrence of this interaction. The calculated Sterm-Volmer constant (*K_SV_*) between the PS and BSA was *K_SV_* = 1.99 × 10^4^ M^−1^, suggesting an effective interaction between TMAP^4+^ and BSA. The *K_SV_* value found for the TMAP^4+^-albumin complex indicated strong significant interaction [26]. In addition, a decrease in TMAP^4+^ absorbance was observed as the BSA concentration increased (Figure 3A). From this graph, the binding constant of PS to the protein was determined using the equation of Benesi–Hildebrand (Equation (1)) [27], plotting the reciprocal of the difference in absorbance at 412 nm caused by the addition of BSA against the reciprocal of the BSA concentration.(1)$$\frac{1}{∆A}=\frac{1}{K_{b}∆\varepsilon \left[{TMAP}^{4+}\right]}.\frac{1}{\left[BSA\right]}+\frac{1}{∆\varepsilon \left[{TMAP}^{4+}\right]}$$

From the relationship between the intersection and the slope of the line in Figure 3B, a K_b_ value of 1.6 × 10^4^ M^−1^ was obtained.

**Figure 2 ijms-26-11267-f002:**
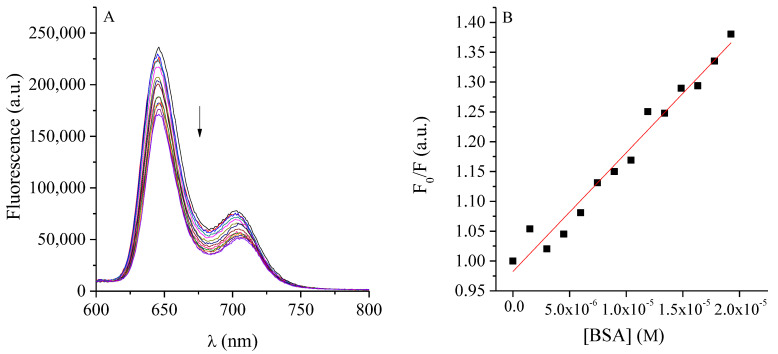
(**A**) Fluorescence spectra of TMAP^4+^ (0.5 μM) titration with BSA (λ_exc_ = 417 nm) in water. (**B**) Stern–Volmer plot for the fluorescence quenching of TMAP^4+^ by BSA in aqueous solution.

**Figure 3 ijms-26-11267-f003:**
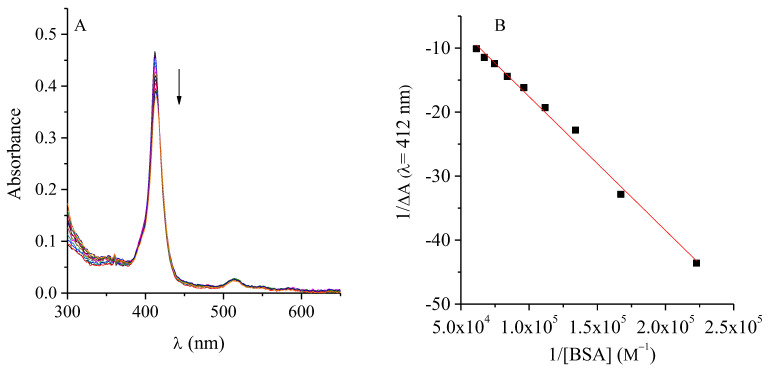
(**A**) Absorbance spectra of TMAP^4+^ (0.5 μM) titration with BSA in PBS. (**B**) Plot of 1/ΔA vs 1/[BSA] in aqueous solution.

Similar behavior has previously been observed in the interaction of cationic PSs with albumin [28,29,30]. As a consequence of this interaction, the photoinactivation of microbial cells could be significantly reduced in the presence of albumin, although photoinactivation could still be significant at longer exposures times. Nitzan et al. reported that high concentration of proteins presents in the medium or serum would prevent photoinactivation of deuteroporphyrin in combination with polymyxin nonapeptide for *Acinetobacter baumannii* strains [31]. They proposed that this may be due to slow incorporation of PS in protein-rich environment, leads to a decrease in the effectiveness of PDI. However, even if it is altered or reduced, the photoinactivation of bacteria is still possible [24,32]. This was also observed in this study, since despite requiring higher concentrations of PS, the photodynamic effect can still be observed in the inactivation of bacteria, being more evident in the Gram-positive strain than in the Gram-negative strain.

### 2.6. TMAP^4+^ and ATBs Growth Curves

Comparing bacterial growth curves in the presence of different antimicrobial agents provides valuable insight into their impact on the population dynamics of the microorganism. The growth profiles of the *S. aureus* cultures treated with both ATBs did not exhibit significant differences compared to the control cultures (Figure 4). In contrast, treatment with TMAP^4+^ resulted in a marked extension of the lag phase, which doubled in duration compared to the controls, requiring approximately 4 h to reach the exponential phase. Similarly, the combination of therapies led to a pronounced prolongation of the lag phase. Specifically, the AMP + TMAP^4+^ combination not only delayed the onset of exponential growth but also reduced the slope of this phase, indicating a slower growth rate. In *S. aureus* cultures treated with the RIF + PS combination, adaptation required more than 6 h before entering the exponential phase. Moreover, CFU/mL counts after 9 h the start of the growth curves reflected a decrease in the number of viable bacteria compared to control cultures (Appendix A).

On the other hand, the growth curve of *E. coli* treated with CFX exhibited similar behavior to that of the control cultures, although it showed a lower slope in the exponential phase and lower absorbance upon reaching the stationary phase (Figure 5). The curve treated with TMAP^4+^ showed an elongation of the latency phase of ~2 h, suggesting that the cells needed more time to adapt to these conditions. Likewise, the cultures treated with PS reached the stationary phase after 7 h, with an absorbance comparable to that of the control cultures. However, the combined treatment of CFX with TMAP^4+^ caused a pronounced delay in growth initiation (~4 h) and lower absorbance upon reaching the stationary phase, consistent with a cumulative antimicrobial effect. These results are confirmed by the CFU/mL count performed 8 h after starting the cultures (Appendix A).

Similarly, Wozniak et al. documented a notable shift in the growth curve as a result of combining PDI, either using a fullerene derivative or Rose Bengal as a PS and different ATBs [33]. Together, these kinetic data reinforce that combining PDI with conventional ATBs enhances overall antimicrobial efficacy, primarily by extending adaptation times and reducing viable populations.

### 2.7. Photostability of ATBs

Photodegradation studies provide valuable insights not only into the stability of the agents, but also into the appropriate treatment conditions [34]. The photostability of the PS was previously evaluated, demonstrating that its photodynamic activity remains stable under the experimental conditions employed [32,34]. Besides, the photostability of the ATBs was assessed under the maximum irradiation conditions used in this study (1 h exposure to white light, with an irradiance of 90 mW·cm^−2^ at room temperature). These experiments aimed to evaluate the stability of the ATBs under the same photo-treatment parameters applied in the MIC, MBC and bacterial growth assays. The extent of photodegradation was monitored by measuring the absorbance intensity as a function of irradiation time in aqueous solution. Representative spectral changes observed after photolysis of the different ATBs are shown in Appendix A. Although AMP is known to be susceptible to hydrolysis under normal conditions [35], both AMP and CFX exhibited no detectable decrease in their characteristic absorption bands after more than 1 h of irradiation, indicating that both molecules remained stable under the tested conditions. Conversely, RIF exhibited a gradual decrease in its 470 nm absorption band (Figure 6), evidencing its photosensitivity. From the plots of ln(A_0_/A) vs. irradiation time (s) of RIF (Figure 7), the photobleaching process was determinate that follows a first-order kinetic behavior, with an observed rate constant of k_obs_ = 1.04 × 10^−4^ s^−1^. The corresponding photodegradation half-life (τ_1/2_) was estimated to be 1.85 h.

These results demonstrate that, among the tested ATBs, only RIF undergoes measurable photodecomposition under the irradiation conditions used, while AMP and CFX remain photostable. Under photoactivation of TMAP^4+^, ROS generated during the photosensitized process, mainly singlet oxygen, may oxidize RIF, leading to its photodecomposition. Such oxidative modifications can alter the ATB structure and contribute to changes in its antimicrobial activity. Nonetheless, the extend RIF degradation is minimal: the calculated τ_1/2_ indicates that less than 27% of the compound is photolyzed during the PDI treatment period. Despite this small photodegradation of the ATB, control experiments demonstrated that this did not result in an appreciable loss of antibacterial efficacy. Moreover, the k_obs_ is substantially lower than values reported for the UV-induced degradation of ceftriaxone [36]. Moreover, quantum yields of photobleaching (Ф_P_) were calculated as [initial rate of disappearance of RIF]/[initial rate of absorption of photons by the reaction mixture] [37]. These results of Ф_P_ for RIF was (1.3 ± 0.2) × 10^−4^, indicates that this ABT undergoes relatively low photodegradation efficiency per absorbed photon. Therefore, the potentiation observed in the combined treatments with PDI and RIF can be attributed to a true additive biological effect. This comparison highlights the relatively high photostability of RIF under the mild visible-light conditions used in this study.

The absence of direct PS–ATB interactions and the photostability of both components confirm that the observed effects stem from true biological synergy rather than chemical artifacts. Moreover, the partial quenching of PS fluorescence by albumin highlights the potential relevance of protein interactions in vivo and suggests that higher PS doses or optimized formulations may be required for systemic applications.

## 3. Materials and Methods

Materials and instrumentation. The ATBs CFX, AMP and RIF and the PS TMAP^4+^ were purchased from Sigma-Aldrich (St Louis, MO, USA). PS stock solutions (0.5 mM) were obtained by dissolution in 1 mL of *N*,*N*-dimethylformamide (DMF) (Merck, Darmstadt, Germany). The concentration was confirmed by spectroscopy (ε = 178144 M^−1^ cm^−1^ at 412 nm) [38]. ATB and enzyme stock solutions were prepared in distilled water, except for RIF, which was dissolved in dimethyl sulfoxide (DMSO) (Merck, Darmstadt, Germany). The concentration of the ATBs stock solution was 1024 μg/mL. Bacterial cultures irradiation was performed on 24 and 96 wells microtiter plates with a Novamat 130 AF (Braun Photo Technik, Nürnberg, Germany) projector equipped with a 150 W halogen lamp. Wavelength range between 350 and 800 nm was selected by optical filters with a fluence rate of 90 mW.cm^−2^ [26]. The absorption of bacterial cultures was determined in a Barnstead Turner SP-830 spectrophotometer (Dubuque, IA, USA) and the light fluence rate was established with a Radiometer Laser Mate-Q, (Coherent, Santa Clara, CA, USA).

Interaction of TMAP^4+^ with ATBs. The interaction between TMAP^4+^ and ATBs was studied by steady-state fluorescence spectroscopy (Horiba Jobin Yvon Inc., Edison, NJ, USA). A TMAP^4+^ solution (0.5 µM) was prepared in phosphate-buffered saline (PBS) and increasing concentrations of each ATB were successively added. After each addition, fluorescence spectra were recorded. The concentration ranges tested were 7.8 × 10^−3^ to 4 µg·mL^−1^ for AMP, 4.8 × 10^−4^ to 0.25 µg·mL^−1^ for RIF, and 6.2 × 10^−2^ to 32 µg·mL^−1^ for CFX. Emission spectra were obtained upon excitation at the *Soret* band maximum of TMAP^4+^ (λ_exc_ = 417 nm), and fluorescence emission was monitored between 600 and 800 nm. The extent of interaction was evaluated by monitoring the decrease in fluorescence intensity (F) relative to the initial fluorescence (F_0_).

Strains and bacterial culture conditions. Gram-positive *Staphylococcus aureus* ATCC 25923 and Gram-negative *Escherichia coli* (EC7) stored in a 10% glycerol stock at −70 °C, were thawed and sub-cultured in Tryptic Soy Broth (TSB) (Britania, Buenos Aires, Argentina) [8,9]. The incubation of *S. aureus* was carried out on an orbital shaker at 100 rpm, while that of *E. coli* under static conditions. Both microbial strains were grown overnight at 37 °C in TSB.

Bacterial inoculum. Overnight cultures of both species were adjusted to an optical density of 0.1 at 625 nm, corresponding to a 0.5 McFarland standard (10^8^ colony-forming units (CFU)/mL) [18]. The inoculum was further diluted in TSB to obtain a final cell concentration of approximately 10^5^ CFU/mL.

MIC and MBC determination. The minimum inhibitory concentration (MIC) and the minimum bactericidal concentration (MBC) were determined using the broth microdilution method in 96-well microplates, according the guidelines of the National Committee for Clinical Laboratory Standards (NCCLS) [19]. For ATB, double serial dilutions of CFX (0.25–128 μg/mL), AMP (0.25–128 μg/mL) and RIF (0.03–128 μg/mL) [39], were prepared in TSB. For the PS, concentration ranges of 0.25 to 32 μM were used. MICs and MBCs were evaluated for ATBs individually and in combination. In contrast to the ATBs, the PS was incubated for 15 min in the dark at 37 °C to promote PS-cell interaction, followed by 30 min of irradiation. In all cases, the standardized inoculum was added, and plates were incubated at 37 °C for 24 h in a humidified chamber. The MIC was defined as the lowest concentration without visible turbidity. For MBC, aliquots were taken from the non-turbid wells, seeded on Tryptic Soy Agar TSA (Britania, Buenos Aires, Argentina) and incubated at 37 °C for 24 h. The MBC was the lowest concentration without bacterial growth. Growth controls (without ATB or PS) and sterility controls (without microorganisms) were included in all determinations, and in the case of PS, dark controls (non-irradiated) were also included.

MIC and MBC of TMAP^4+^ + ATB. Serial two-fold ATB dilutions were prepared, followed by addition of TMAP^4+^ (either at a fixed concentration corresponding to its predetermined MIC or at varying levels) and the microbial inoculum. Plates were incubated in the dark for 15 min, followed by 30 min of irradiation. Subsequently, the microplates were incubated 24 h at 37 °C in a humid chamber. After incubation, both MIC and MBC were determined as previously described.

Interaction of TMAP^4+^ with BSA. The interaction between TMAP^4+^ and bovine serum albumin (BSA) (Sigma-Aldrich, St Louis, MO, USA) was investigated by steady-state fluorescence spectroscopy under similar experimental conditions to those described for the ATB interaction assays [30]. Stock solution of BSA (1.5 mM) was prepared in water. A TMAP^4+^ solution (0.5 µM) was prepared in PBS and increasing concentrations of BSA (1.5 × 10^−6^ to 1.9 × 10^−5^ M) were successively added to the porphyrin solution. Fluorescence emission spectra were recorded (λ_exc_ = 417 nm.) and emission was monitored in the 600–800 nm range. Changes in F relative to F_0_ was analyzed as a function of BSA concentration to evaluate the binding interaction between the protein and TMAP^4+^. To analyze the data from the quenching experiments, the Stern–Volmer plot was performed according to Equation (2)F_0_/F = 1 + K_SV_ [BSA](2)
where F and F_0_ are the fluorescence intensities of TMAP^4+^ in the presence and absence of BSA, respectively. K_SV_ represents the Stern–Volmer quenching constant.

Growth curves. A 20 μL aliquot from overnight cultures of *S. aureus* or *E. coli* was aseptically transferred to 20 mL of fresh TSB. Aliquots (2 mL) from this suspension were distributed into sterile test tubes and the MIC of each ATB was added. The cultures were irradiated for 1 h and then incubated at 37 °C with shaking until stationary phase was reached. Absorbance at 660 nm was measured every 30 min using 1 cm path length cuvettes. This wavelength was selected to avoid interference with the porphyrin absorption bands. At the end of the incubation period (8 h), ten-fold serial dilutions were performed in PBS from each sample. The dilutions were spread in triplicate on TSA plates and incubated at 37 °C for 24 h. Finally, the number of colonies formed were counted and the results were expressed in CFU/mL.

Photobleaching evaluation. Photobleaching experiments were conducted to assess the light-induced degradation of the ATB. Fresh aqueous solutions of each ATB were prepared immediately before to irradiation (AMP 2 mg·mL^−1^ in water, RIF 8 µg·mL^−1^ in DMSO/H_2_O) and exposed to white light at an irradiance of 90 mW·cm^−2^ for 60 min at room temperature. The degradation progress was monitored by recording the absorbance spectra using a UV-Vis spectrophotometer (Shimadzu Corporation, Tokyo, Japan). For RIF, the photodegradation kinetics were analyzed by plotting ln(A_0_/A) vs. time, where A_0_ represent the initial absorbance and time-dependent absorbance values, respectively. The observed photobleaching rate constants (k_obs_^P^) were obtained from these semilogarithmic plot and the photodegradation lifetime (τ^P^) was calculated from ln 2/k_obs_^P^ [40].

Statistical analysis. Each experiment was repeated separately three times, containing each test in quintuplicate. The amount of DMF (<1% *v*/*v*) used in each experiment was not toxic to microbial cells. Data were depicted as the mean±standard deviation of each group. Variation between each experiment was calculated using the one-way ANOVA, with a confidence level of 95% (*p* < 0.05) considered statistically significant, using OriginPro v9.0 2023 (OriginLab Corporation, Northampton, MA, USA).

## 4. Conclusions

The present study provides a comprehensive analysis of the interactions and combined antimicrobial effects between the cationic PS TMAP^4+^ and ATBs with distinct mechanisms of action. Spectroscopic studies revealed no detectable physical interactions between TMAP^4+^ and any of the ATBs tested, indicating that their simultaneous application does not lead to mutual quenching or loss of photophysical activity. Photostability assays further demonstrated that AMP and CFX are stable under the irradiation conditions used for PDI, whereas RIF exhibited only minimal photodecomposition during treatment, confirming that light exposure does not compromise its antimicrobial efficacy.

Moreover, fluorescence quenching studies with bovine serum albumin (BSA) revealed that TMAP^4+^ interacts strongly with proteins, which could influence PS bioavailability in complex biological environments. Although such interactions may partially reduce photoinactivation efficiency, the persistence of significant antimicrobial activity suggests that PDI remains effective even in protein-rich media.

When combined with ATBs, PDI produced a reduction in MIC values compared to treatments with each agent alone, evidencing additive effects. In *S. aureus*, the combinations of TMAP^4+^ with AMP or RIF led to significant post-irradiation decreases in both MIC and MBC, confirming that the ROS generated during photoactivation potentiate ATB action. In *E. coli*, the combination of TMAP^4+^ with CFX also resulted in enhanced antibacterial activity, though the effect was less pronounced, reinforcing the notion that PS penetration and outer membrane permeability are critical parameters for Gram-negative species. Growth kinetics analyses further supported the additive nature of the combined treatments, showing extended lag phases and reduced viable cell counts in bacterial populations exposed to PDI–ATB combinations compared to individual agents.

Overall, these findings demonstrate that PDI mediated by TMAP^4+^ enhances the antimicrobial performance of conventional ATBs through complementary mechanisms, without compromising drug stability. The combination of PDI with ATBs such as AMP, RIF, or CFX offers a promising strategy to potentiate antimicrobial efficacy, minimize required drug concentration, and potentially mitigate the development of resistance.

## Figures and Tables

**Figure 1 ijms-26-11267-f001:**
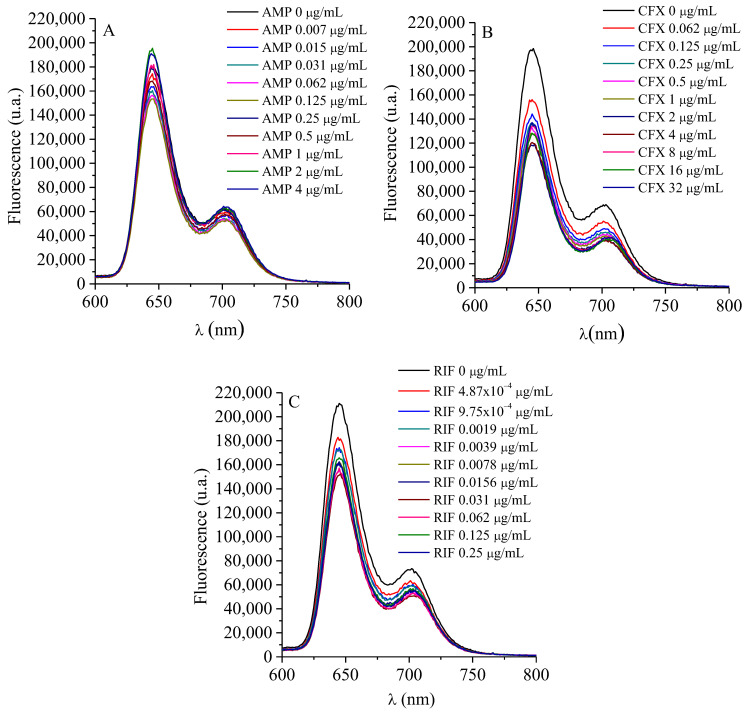
Fluorescence spectra of TMAP^4+^ (0.5 µM) titration with (**A**) AMP, (**B**) RIF and (**C**) CFX in water (λ_exc_ = 417 nm).

**Figure 4 ijms-26-11267-f004:**
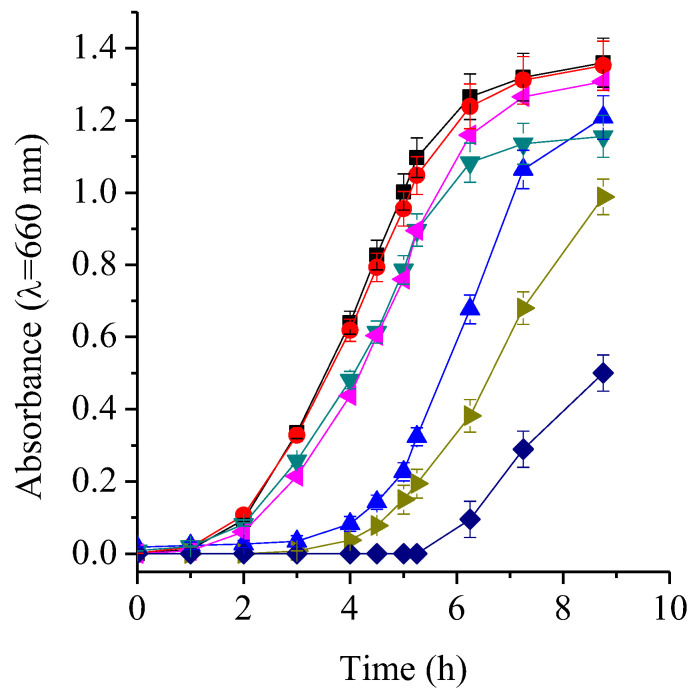
Growth curves of *S. aureus* cells with 0.0156 μg/mL AMP (▼), 0.0039 μg/mL RIF (◀), 0.5 μM TMAP^4+^ (▲), 0.0156 μg/mL AMP + 0.5 μM TMAP^4+^ (▶), 0.0039 μg/mL RIF + 0.5 μM TMAP^4+^ (◆) and irradiated for 1 h with visible light in TSB at 37 °C. Untreated control culture irradiated (⯀) and culture with 0.5 μM TMAP^4+^ in dark (●). Values represent mean ± standard deviation of three separate experiments.

**Figure 5 ijms-26-11267-f005:**
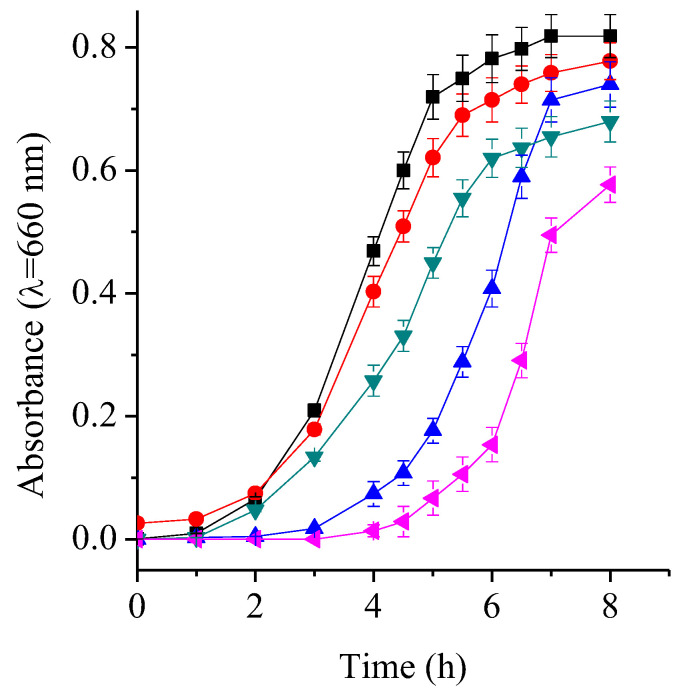
Growth curves of *E. coli* cells with 4 μg/mL CFX (▼), 16 μM TMAP^4+^ (▲), 4 μg/mL CFX + 16 μM TMAP^4+^ (◀) and irradiated for 1 h with visible light in TSB at 37 °C. Untreated control culture irradiated (⯀) and culture with 16 μM TMAP^4+^ in dark (●). Values represent mean ± standard deviation of three separate experiments.

**Figure 6 ijms-26-11267-f006:**
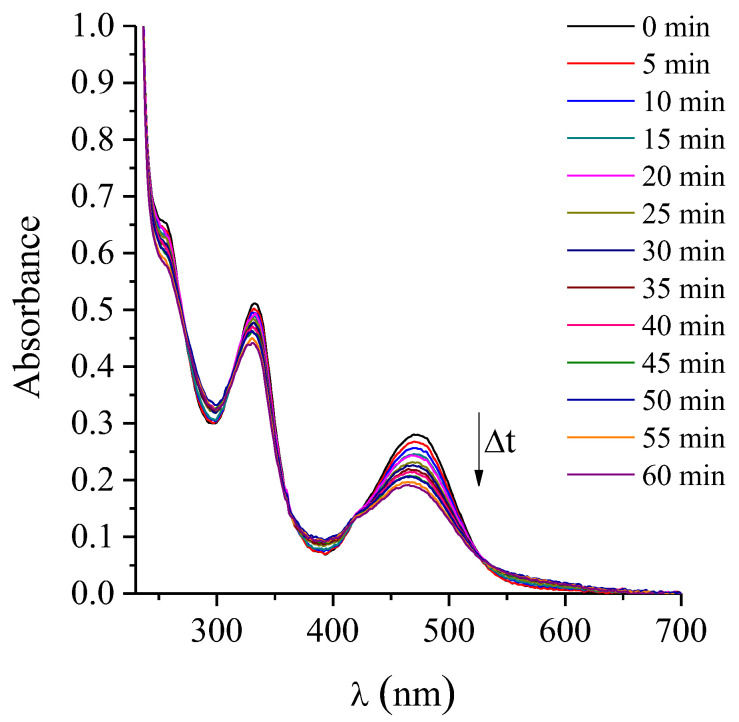
Variation of UV–vis absorption spectra of RIF (8 μg/mL) after different irradiation times with an irradiance of 90 mW·cm^−2^ at room temperature in water.

**Figure 7 ijms-26-11267-f007:**
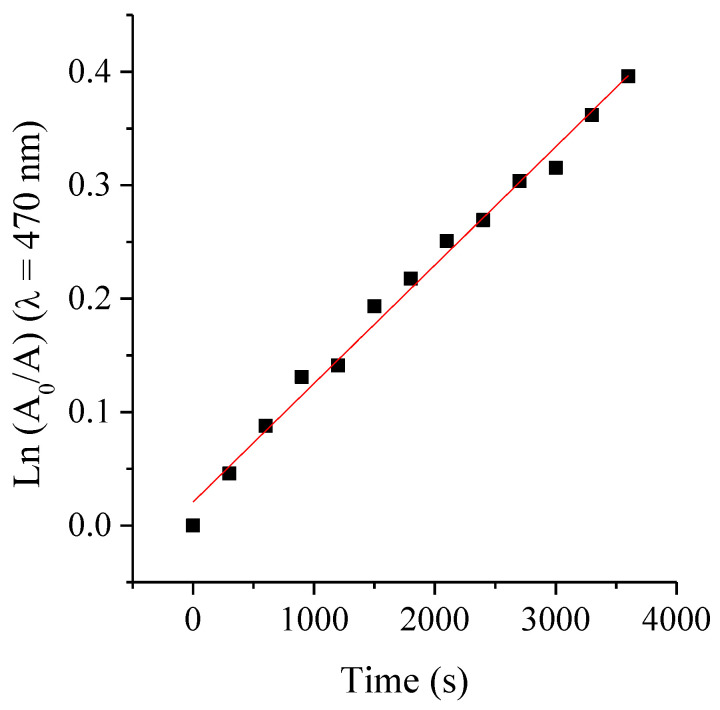
Plots of ln(A_0_/A) at 471 nm of RIF against the irradiation time (s), fitted by the pseudo-first-order model.

**Table 1 ijms-26-11267-t001:** Determination of MIC and MBC of TMAP^4+^ (μM) for *S. aureus* and *E. coli*.

Strains	MIC in Dark	MIC Post PDI	MBC in Dark	MBC Post PDI
*S. aureus*	32	0.5	32	16
*E. coli*	32	32	32	32

**Table 2 ijms-26-11267-t002:** Determination of MIC and MBC of TMAP^4+^ (μM) + ATB (μg/mL) at a fixed PS concentration for *S. aureus* and *E. coli*.

Strain	Antimicrobial Agents	MIC in Dark	MIC Post PDI	MBC in Dark	MBC Post PDI
*S. aureus*	AMP + TMAP^4+^	0.1250 + 0.5	0.0156 + 0.5	0.2500 + 0.5	0.1250 + 0.5
	RIF + TMAP^4+^	0.0156 + 0.5	0.0039 + 0.5	0.0625–0.0312 + 0.5	0.0312 + 0.5
*E. coli*	CFX + TMAP^4+^	<0.1250 + 32	<0.1250 + 32	<0.1250 + 32	<0.1250 + 32
	CFX + TMAP^4+^	4 + 16	4 + 16	16 + 16	8 + 16

**Table 3 ijms-26-11267-t003:** Determination of MIC and MBC of TMAP^4+^ (μM) + ATB (μg/mL) at different PS concentrations for *S. aureus* and *E. coli*.

Strain	Antimicrobial Agents	MIC in Dark	MIC Post PDI	MBC in Dark	MBC Post PDI
*S. aureus*	AMP + TMAP^4+^	0.2500 + 2	0.0156–0.0039 + 0.1250–0.0039	0.2500 + 2	0.1250 + 1
	RIF + TMAP^4+^	0.0156–0.0078 + 1–0.5	0.0039–0.0009 + 0.2500–0.0625	0.0625–0.0312 + 4–2	0.0625–0.0312 + 4–2
*E. coli*	CFX + TMAP^4+^	2–1 + 8–4	2–1 + 8–4	8 + 32	8 + 32

## Data Availability

The original contributions presented in this study are included in the article/Appendix A. Further inquiries can be directed to the corresponding author.

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
