# Peer review of "Photodynamic Inactivation Enhances Antibiotic Efficacy Without Affecting Drug Stability: Insights into Photosensitizer–Antibiotic Combination Therapies"

_ijms, 2025, doi:10.3390/ijms262311267_

Round 1

Reviewer 1 Report

Comments and Suggestions for Authors

I reviewed the manuscript entitled: “Photodynamic inactivation enhances antibiotic efficacy with-2 out affecting drug stability: Insights into photosensitizer–anti-3 biotic combination therapies” International Journal of Molecular Sciences (Manuscript ID: ijms-3965311). This paper describes the photodynamic inactivation effect of 5,10,15,20-tetrakis(4-N,N,N-trime-53 thylammoniumphenyl) porphyrin (TMAP4⁺) with ATPs. The important findings of this study is a synergy effect of porphyrin and ATBs combination. I think that this is an interesting study. However, several issues should be considered and resolved before the publication.

Noticed points
1)    Introduction, line 52: The singlet oxygen generation is mentioned. The quantum yield of singlet oxygen generation by TMAP4⁺ should be provided and discussed.
2)    The dark toxicity of TMAP4⁺ without ATBs is mentioned (lines 107~109). The mechanism of dark toxicity of TMAP4⁺ is interesting information for readers.
3)    The PS concentration dependent MIC and MBC was examined (Tables 1 and 2). The data plot may be helpful to understand the PS concentration effect.
4)    Interaction between porphyrin and BSA: Fluorescence spectral change showed qualitatively the interaction between TMAP4⁺ and BSA. However, the quantitative evaluation is difficult from the Stern-Volmer constant. The fluorescence intensity decrease may be due to the absorption spectrum change (decrease) through the binding interaction of TMAP4⁺ with BSA. The absorption spectral change should be provided. The binding constant, which can be estimated from the absorption spectral change, is more appropriate to evaluate the interaction.
5)    About the Photostability of ATBs: The photodecomposition depends on the intensity of light (fluence of photon). The condition, white light at an irradiance of 90 mW·cm⁻² for 60 min at room temperature, should be provided in the text and Figure caption. To evaluate the photostability, the photodecomposition quantum yield is more appropriate.
6)    A brief explanation about the photodecomposition mechanism of RIF, for example, an oxidation by singlet oxygen produced from the photosensitized reaction of TMAP4⁺, may be important information.

Reviewer 2 Report

Comments and Suggestions for Authors

Dear Authors,

I have reviewed your manuscript, titled "Photodynamic inactivation enhances antibiotic efficacy without affecting drug stability: Insights into photosensitizer–antibiotic combination therapies" and concluded that it is an interesting study dealing with a topic relevant to the journal's readership, International Journal of Molecular Sciences. I think this work is suitable and within the scope of the International Journal of Molecular Sciences. However, some issues need to be addressed.

  1. You stated that the AMP and CFX remained photostable, is this correct? Since AMP is not even stable under normal conditions (it is succseptible to hydrolysis)?
  2. In introduction section I suggest to add one more paragraph findings about am-picillin, rifampicin, andcephalexin.
  3. For Figure 1. Fluorescence spectra of TMAP4+ (0.5 μM) titration with (A) AMP, (B) RIF and (C) CFX in water (exc= 417 nm) kinetic curves should be presented as inset as well as r values.
  4. Given that RIF exhibited partial photodegradation under the applied light conditions, how might this degradation influence its antimicrobial activity during the PDI experiments?
  5. The manuscript would benefit from specifying the software and version employed for the statistical analysis, as this information is important for ensuring reproducibility.
  6. Could the authors clarify the wavelength and optical path length used for measuring absorbance in the bacterial growth curves? This information would help in comparing growth kinetics and determining whether optical density measurements directly correspond to CFU/mL values.
  7. Add more information in the conclusion sections to sum up the findings of this work.

Round 2

Reviewer 1 Report

Comments and Suggestions for Authors

I checked the revised form. I feel almost issues have been resolved. Please consider the following two points.

1)    About the original comment 5): If possible, the determination of ATBs photodecomposition quantum yield should improve this paper.
2)    About the original comment 6): Page 11, line 33: “ROS” means singlet oxygen?
